# Pan-Genome Analysis of *Staphylococcus aureus* Reveals Key Factors Influencing Genomic Plasticity

Na Liu,[a,b,c] Danping Liu,[a,b,c] Kexin Li,[f] ⬤ Songnian Hu,[d,e] Zilong He[a,b,c]

[a]School of Engineering Medicine, Beihang University, Beijing, China
[b]Key Laboratory of Biomechanics and Mechanobiology, Beihang University, Ministry of Education, Beijing, China
[c]Key Laboratory of Big Data-Based Precision Medicine, Beihang University, Ministry of Industry and Information Technology of the People's Republic of China, Beijing, China
[d]State Key Laboratory of Microbial Resources, Institute of Microbiology, Chinese Academy of Sciences, Beijing, China
[e]University of Chinese Academy of Sciences, Beijing, China
[f]Systems Biology and Bioinformatics (SBI), Leibniz Institute for Natural Product Research and Infection Biology—Hans Knöll Institute (HKI), Jena, Germany

**ABSTRACT** The massive quantities of bacterial genomic data being generated have facilitated in-depth analyses of bacteria for pan-genomic studies. However, the pan-genome compositions of one species differed significantly between different studies, so we used *Staphylococcus aureus* as a model organism to explore the influences driving bacterial pan-genome composition. We selected a series of diverse strains for pan-genomic analysis to explore the pan-genomic composition of *S. aureus* at the species level and the actual contribution of influencing factors (sequence type [ST], source of isolation, country of isolation, and date of collection) to pan-genome composition. We found that the distribution of core genes in bacterial populations restrained under different conditions differed significantly and showed "local core gene regions" in the same ST. Therefore, we propose that ST may be a key factor driving the dynamic distribution of bacterial genomes and that phylogenetic analyses using whole-genome alignment are no longer appropriate in populations containing multiple ST strains. Pan-genomic analysis showed that some of the housekeeping genes of multilocus sequence typing (MLST) are carried at less than 60% in *S. aureus* strains. Consequently, we propose a new set of marker genes for the classification of *S. aureus*, which provides a reference for finding a new set of housekeeping genes to apply to MLST. In this study, we explored the role of driving factors influencing pan-genome composition, providing new insights into the study of bacterial pan-genomes.

**IMPORTANCE** We sought to explore the impact of driving factors influencing pan-genome composition using *Staphylococcus aureus* as a model organism to provide new insights for the study of bacterial pan-genomes. We believe that the sequence type (ST) of the strains under consideration plays a significant role in the dynamic distribution of bacterial genes. Our findings indicate that there are a certain number of essential genes in *Staphylococcus aureus*; however, the number of core genes is not as high as previously thought. The new classification method proposed herein suggests that a new set of housekeeping genes more suitable for *Staphylococcus aureus* must be identified to improve the current classification status of this species.

**KEYWORDS** pan-genome, genomic plasticity, sequence type, core gene, housekeeping gene

**Ad Hoc Peer Reviewer** ⬤ Joao Carlos Gomes-Neto, University of Nebraska-Lincoln

Address correspondence to Zilong He, hezilong@buaa.edu.cn.

The authors declare no conflict of interest.

The term pan-genome was first proposed by Tettelin et al. in their genomic study of *Streptococcus agalactiae* in 2005, defining it as the entirety of genomic information within one species (1). Numerous strains cause changes in the gene content of bacterial genomes through gene loss, gene duplication, and horizontal gene transfer, resulting in plasticity of the genome (2, 3). The alternation of genes in the genome may play a role in

adaptation to specific growth conditions, including those involving symbiosis, host-cell interactions, and pathogenicity (4). Genomic plasticity may lead to different environmental adaptations in the strains within a species—e.g., the adaptive dominance genes in *Bacillus mycoides* that allow these organisms to inhabit different ecological niches as a result of adaptive evolution (5, 6).

There have been several studies demonstrating the links between genomic plasticity and phenotype. Sahl et al. found phenotype-specific genes in *Acinetobacter baumannii* strains isolated from human anus, blood, and wound samples but found that the specific genes were not restricted to one phenotype when more isolates were included for each phenotype (7). In a comparative genomics study of *Enterococcus faecalis*, Zhong et al. found that five dairy-specific genes, possibly constituting a complete lactose metabolism pathway (*lacF*, *lacA/B*, *lacD*, *lacG*, and *lacC*), were present in almost all dairy isolates, demonstrating the active role of the environment in shaping the genome of *E. faecalis* (8). Jia et al. found that *Acinetobacter johnsonii* had an open pan-genome and that clinically sourced strains contained more genes associated with translational modifications, $\beta$-lactamase, and defense mechanisms, while environmentally sourced strains accumulated more genes associated with material degradation (9). Horesh et al. implied that pan-genomic studies should focus on the influence of lineages and proposed a population structure-aware pan-genomic classification approach through which they revealed a unique pattern of evolutionary dynamics for 7,500 *Escherichia coli* genomes (10). However, because of the small number and low diversity of strains included in most previous studies, the pan-genome situation at the species level has not been fully demonstrated. Moreover, factors influencing bacterial genome plasticity, such as source of isolation, date of collection, and lineage-related differences, have not been systematically compared.

With the rapid development of sequencing technologies and the reduced sequencing costs in recent years, massive quantities of bacterial genomic data have been published, making it feasible to analyze bacteria in depth for pan-genomic studies. In this study, we used *Staphylococcus aureus* as a model organism for studying the pan-genome. We downloaded sequencing data for 5,217 strains of *S. aureus* from the NCBI database, screened the strains by sequence type (ST), and assessed the pan-genome composition of *S. aureus* based on the strains obtained from the screening. Meanwhile, we analyzed the driving factors that may influence bacterial genetic dynamics—ST, source of isolation, country of isolation (geographical location), and date of collection—and assessed the role of these factors in influencing gene distribution. We aimed to explore the importance of the role played by driving factors in influencing pan-genome composition to provide new insights into studying bacterial pan-genomes.

## RESULTS

**Large-scale diversity assessment of *Staphylococcus aureus* core genes.** We performed multilocus sequence typing (MLST) identification on 5,217 *S. aureus* strains available in the NCBI database. According to statistical results, 5,001 strains were identified as known types and 216 strains as unknown types (see Data Set S1 in the supplemental material). We selected 1,345 strains with known STs and 216 strains with unknown STs for pan-genome analysis. (Forty-two strains were deleted in the process of calculating the core genes; therefore, a total of 1,519 strains were used.) It can be seen from the line chart that the number of core genes calculated based on the coexistence of ≥95% of the strains was ~1,000 (Fig. 1A). The number of core genes calculated based on their coexistence in ≥99% of the strains decreased significantly with an increasing number of genomes when there were fewer than 300 genomes. However, the number of core genes stabilized and remained at ~500 with increasing sample quantity. The number of core genes based on coexistence in 100% of the strains decreased with an increasing number of genomes. Although the decrease in the number of core genes slowed as the number of extracted genomes increased, a few genes were still continuously excluded from the ranks of core genes. Moreover, although there were 162 genes in the core genes of 1,519 strains (100%), 61 genes remained that coded for hypothetical proteins with unknown functions (Fig. 1B).

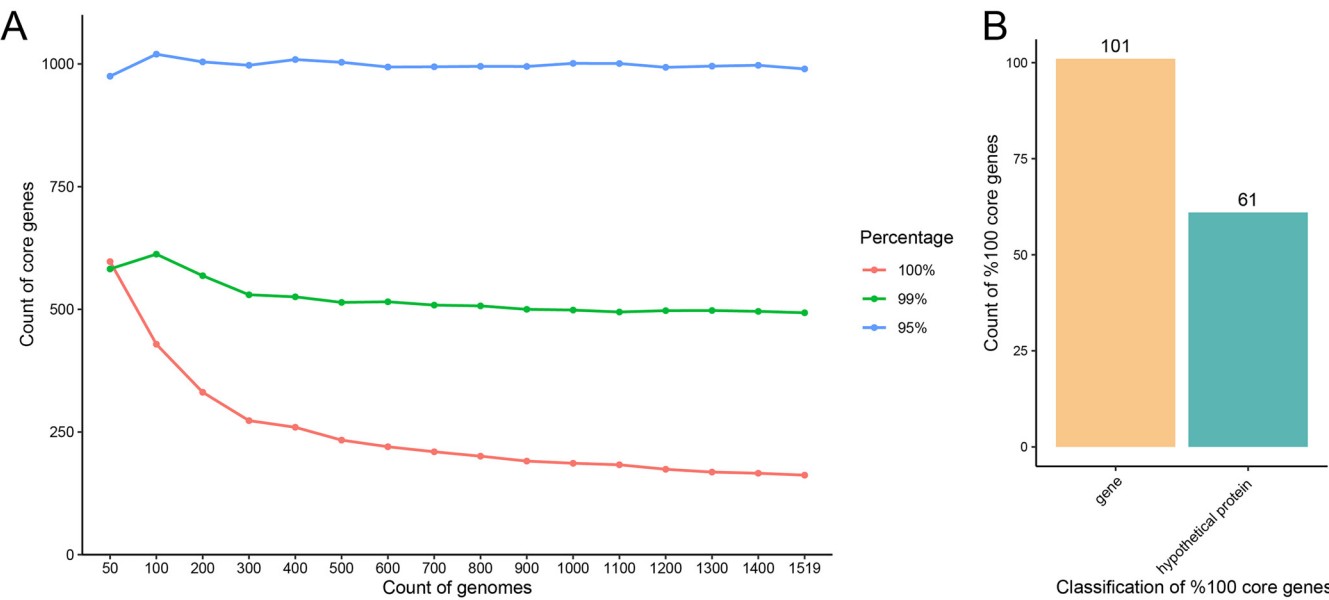

**FIG 1** Distribution of core genes in 1,519 *S. aureus* strains. (A) Pan-genomic analysis of strain populations with different genomic numbers by random sampling. The 95 and 99% lines entered a plateau, while the 100% line has been on a downward trend. (B) Annotation of gene functions in core genes (100%) in 1,519 *S. aureus* strains with genes coding for proteins of known function and putative proteins.

**Dynamic composition of core genomes in different populations.** To explore the influence of related factors on the distribution of bacterial core genes, we screened 2,454 strains for which ST and other important metainformation, such as source of isolation, were available (Data Set S3). According to the statistics (Fig. S1), we selected strains and grouped them according to four observations: (i) different types, including ST8 strains ($n = 678$), ST22 strains ($n = 298$), and ST5 strains ($n = 326$); (ii) different types isolated from blood, including 50 strains each of ST8_blood, ST22_blood, and ST5_blood; (iii) ST8 strains that were isolated from different parts, including 50 strains each of ST8_blood, ST8_nasal, and ST8_soft tissue; and (iv) ST8_blood strains that were isolated in different years, including ST8_blood_2015 ($n = 50$), ST8_blood_2016 ($n = 41$), and ST8_blood_2017 ($n = 27$). We found that each ST strain had a certain number of core genes (95%) unique to that ST, indicating that some core genes were dynamically distributed among different ST strains (Fig. 2A). Strains with different STs still exhibited the dynamics of core genes when we limited the source factor (Fig. 2B). After limiting the strains to ST8, we found that although the distribution of core genes of strains at different sources was relatively concentrated (the number of intersections of the three increased), it was obvious that the core genes among strains at different sources were also dynamically distributed (Fig. 2C). Furthermore, when we restricted ST8 and blood to observe strains isolated in different years, we yet again observed characteristics related to the dynamic distribution of core genes (Fig. 2D).

**Sequence types may drive the dynamic distribution of the pan-genome.** We further screened the 2,454 strains used in the above analysis, selecting a total of 1,519 strains for which ST and source of isolation, country, and year information were available (111 STs, 14 isolation sources, 30 countries, and 23 collection years) (Data Set S3). Based on the presence of 16,794 genes in all strains, correlation analysis was performed on the number of strains ($y$) with a particular core gene that identified a high linear correlation between $y$ and the variables $x_1$ to $x_4$: the number of STs with the gene present ($x_1$), the number of sources with the gene present ($x_2$), the number of countries with the gene present ($x_3$), and the number of years during which the gene was present ($x_4$) ($r = 0.966237359933361 > 0.8$). The resulting multiple-linear-regression equation is as follows:

$$y = -24.339335999024144 + 14.16695064x_1 - 5.60088706x_2 + 0.72108106x_3 + 0.86291475x_4$$

All four variables were significantly correlated with the number of strains containing the gene, while $x_3$ and $x_4$ had little effect. In contrast, ST and the number

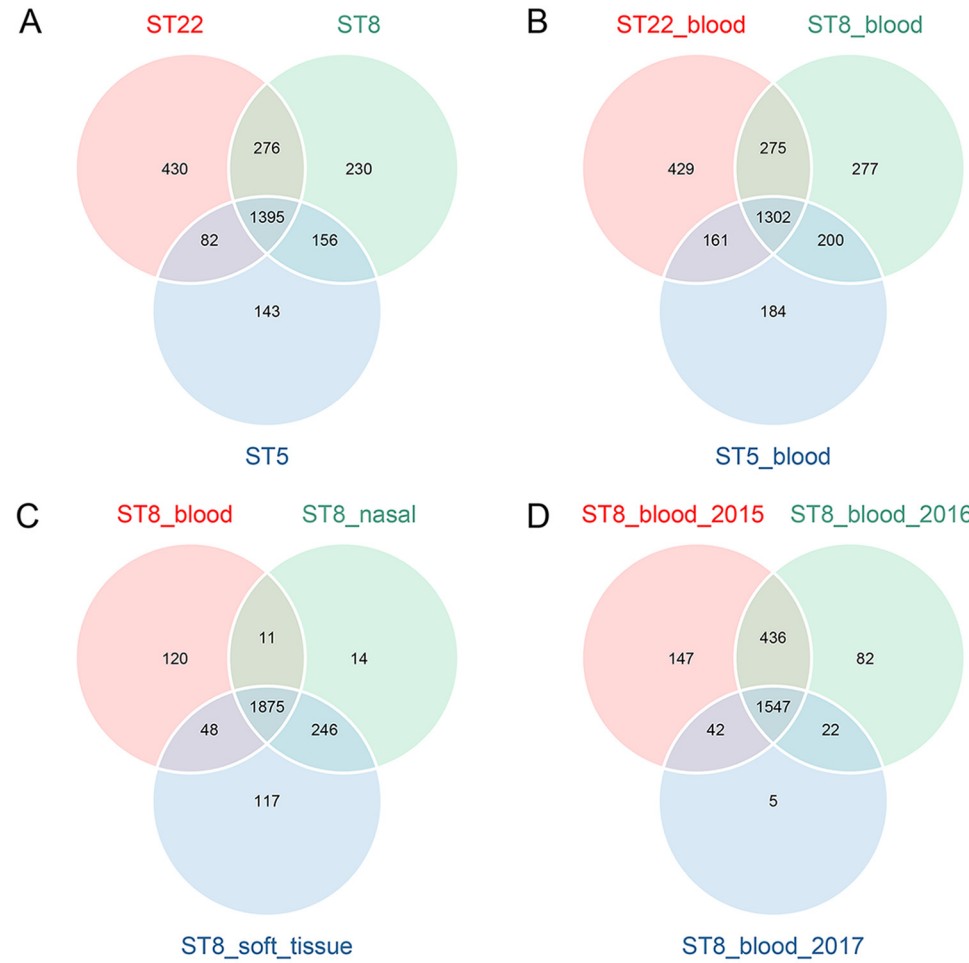

**FIG 2** Venn diagrams of core gene distribution for different groupings of strains. (A) Distribution of core genes for strains of ST8 ($n = 678$), ST22 ($n = 298$), and ST5 ($n = 326$). Labels show the number of genes in that section. (B) Fifty strains each of ST8, ST22, and ST5 isolated from blood; (C) 50 ST8 strains each isolated from blood, soft tissue, and nasal cavity; (D) ST8 strains collected from blood in 2015 ($n = 50$), 2016 ($n = 41$), and 2017 ($n = 27$).

of sources ($x_1$ and $x_2$) both significantly affected the number of strains with the gene.

When evaluating the dynamics of all genes present in 1,519 strains, we found that in addition to the genes present in all strains clustered into a "core gene region," genes present in only certain strains clustered to form a "local core gene region." Most of these local core gene region strains had the same ST, and the size of these local core gene regions cannot be ignored. We speculate that the same ST strains have greater conservation in gene distribution (Fig. S2). To show the relationship between strain clustering and strain-related information more clearly, we selected 5 STs and then selected 5 strains from 5 sources of isolation in each ST; we should note that it is possible that partially typed strains did not entirely contain these 5 sources. A total of 41 strains were further analyzed. We found that strains clustered by ST even when the number of strains was minimal and metainformation was abundant. Furthermore, we found local core gene regions within the same ST (Fig. 3). Among them, ST5 and ST105 clustered, and the distinction between the local core gene regions of the two types was not apparent, with the local core gene regions of the two types likely being shared (probably because the two types have only a few SNP sites between housekeeping genes). In the principal-component clustering of the shell gene, we selected 4 STs that have local core gene regions in Fig. S2 as the labels of the strains (Fig. 4A). We found that ST8 in cluster 2, ST22 in cluster 3, and "Other STs" in cluster 3 corresponded to each other (Fig. 4B). Both ST5 and ST105 were located in cluster 1, which was related to the similar distribution

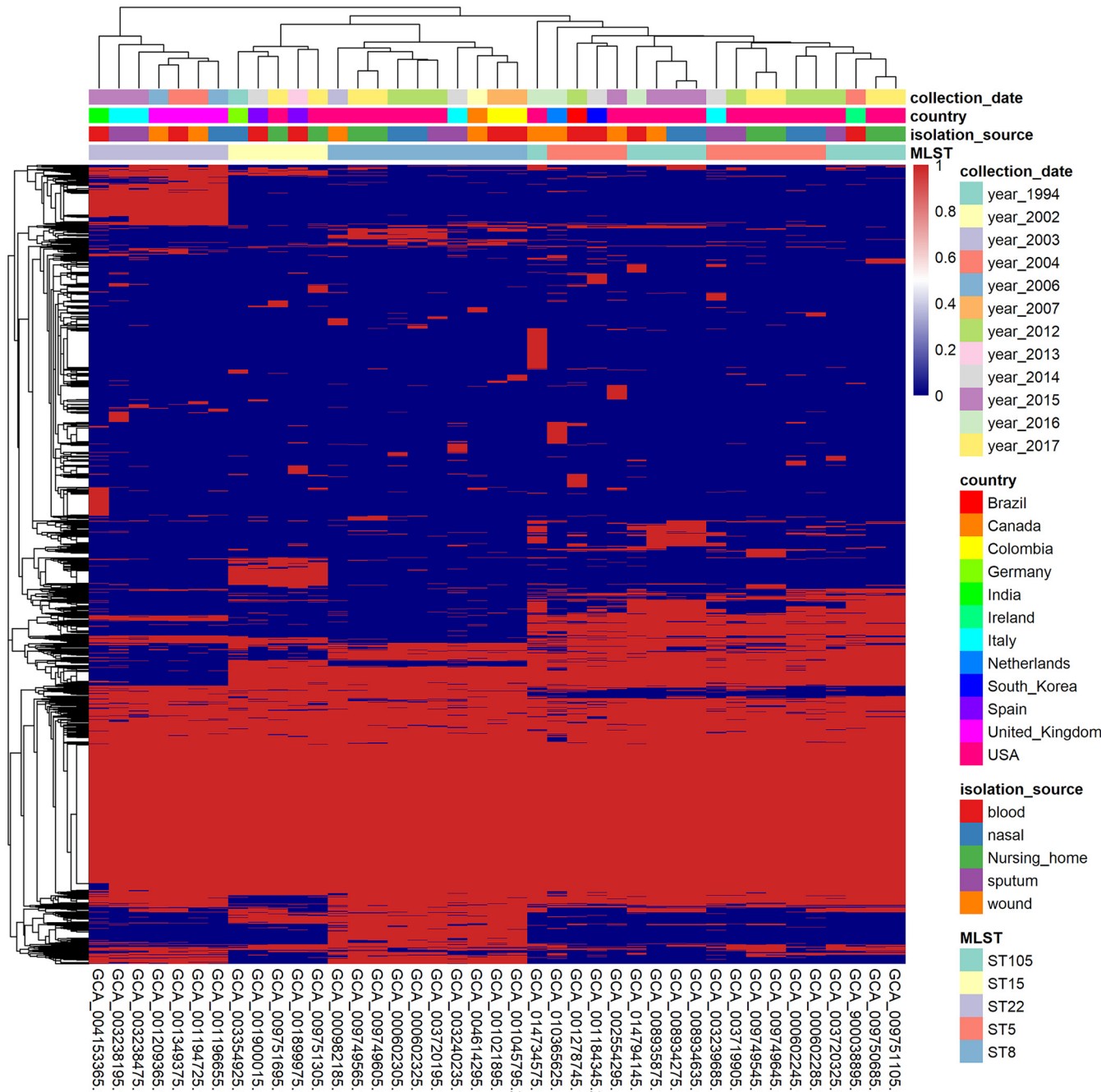

**FIG 3** Heat map of the gene-strain matrix. All genes in the 41 *S. aureus* strains were clustered using hierarchical clustering. Strains with the same ST were clustered, and "local core gene regions" emerged.

of local core gene regions of ST5 and ST105 as mentioned before. In addition, a small number of strains of the "Other STs" are also located in clusters 1, 2, and 3. The results of this analysis led us to believe that ST plays a significant role in the dynamic distribution of bacterial genes.

**Effects of genome plasticity on whole-genome alignments in phylogenetic tree construction.** According to the statistical single nucleotide polymorphism (SNP) results in the five groups, the distribution of SNPs in the three groups restricted to ST8 was similar, with ~25% of SNPs located in noncoding regions (Fig. 5). In addition, a small number of SNP sites were located in the dispensable gene region; these were relatively few in terms of number and proportion. However, when we removed the restriction

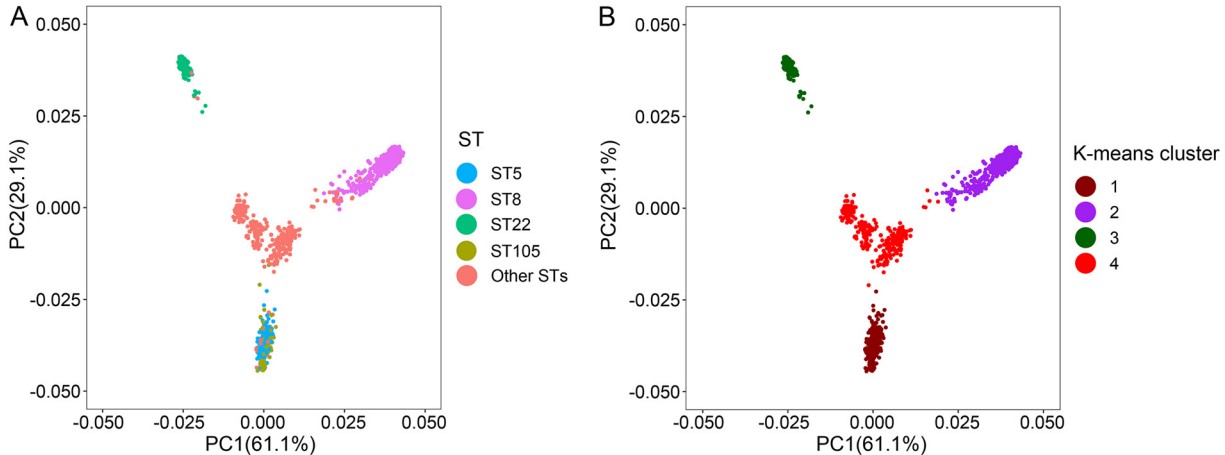

**FIG 4** Clustering analysis of shell genes to show the key role of ST in genomic plasticity. The presence and absence of shell genes in each strain are indicated by 0 and 1 based on the results of the pan-genome analysis of the strains. The binary data matrix was transformed into a Jaccard distance matrix and subjected to principal-component analysis, and two principal components were taken as input. (A) The horizontal and vertical coordinates are the load matrix coefficients of the two principal components, and the strains are labeled with ST to show the clustering of the strains in the different STs. (B) The two principal components are clustered into 4 clusters using K-means.

that the strain under consideration was ST8, the total number of SNPs exhibited a change in the order of magnitude. The proportion of SNPs located in the dispensable gene region also increased significantly. Moreover, the number of SNPs in the species-level grouping increased more than those in groupings isolated from blood and the proportion of SNPs located in the dispensable gene region also increased significantly, accounting for nearly 18% of all SNPs. Phylogenetic analysis based on whole-genome alignment is a commonly used method for screening and epidemic analysis of a species in a geographical area. The SNPs in the dispensable gene region are likely to interfere with the phylogenetic analysis and affect the reliability of the results.

**New classification strategy for *Staphylococcus aureus*.** Calculations based on the results of the pan-genomic analysis software Roary showed that 34% of the 5,217 *S. aureus* strains carried the housekeeping gene *tpi*, and the frequency of detection of *aroE* and *glpF* was less than 60% (Fig. 6A). Based on the core gene distribution of 1,519 strains, we screened 10 new marker genes from 101 core genes with known functions (Table 1). The frequency of detection of the new marker genes in 5,217 strains of *S. aureus* was over 90% (Fig. 6A). Furthermore, we additionally downloaded 5,289 strains of *S. aureus* (Data Set S4) for validation and found that the detection rate of the housekeeping gene *arcC* in the MLST was 24%, while the detection frequencies of the newly screened marker genes were all more than 95% (Fig. S3).

The 10 genes identified after screening were subjected to multiple-sequence alignment, and 1,519 strains of *S. aureus* were divided into 239 groups. This new classification method is also a nucleic acid sequence-based bacterial typing method. However, we reselected 10 genes that aligned more appropriately with the current species of *S. aureus* under analysis and allowed the classification of 216 strains of this study that could not be identified by MLST. MLST identification is often used for epidemiological monitoring and research involving the evolution of bacteria; therefore, we selected 3 strains from each of the 7 STs for further phylogenetic analysis (Fig. 6B). The two methods grouped strains in approximately the same way but differed somewhat in the branches. The ST5 and ST105 strains belonged to group 111, and the two types of housekeeping genes only have SNP sites between *yqiL*, while the ST22, ST45, and ST121 strains were divided into two groups containing only one or two new alleles that had SNP sites. Furthermore, the ST1 and ST8 strains were grouped in the same way as groups 91 and 161.

## DISCUSSION

In this study, strains with core genes at 95% and 99% thresholds in the highly diverse *S. aureus* population finally entered the plateau phase, indicating that there are indeed a

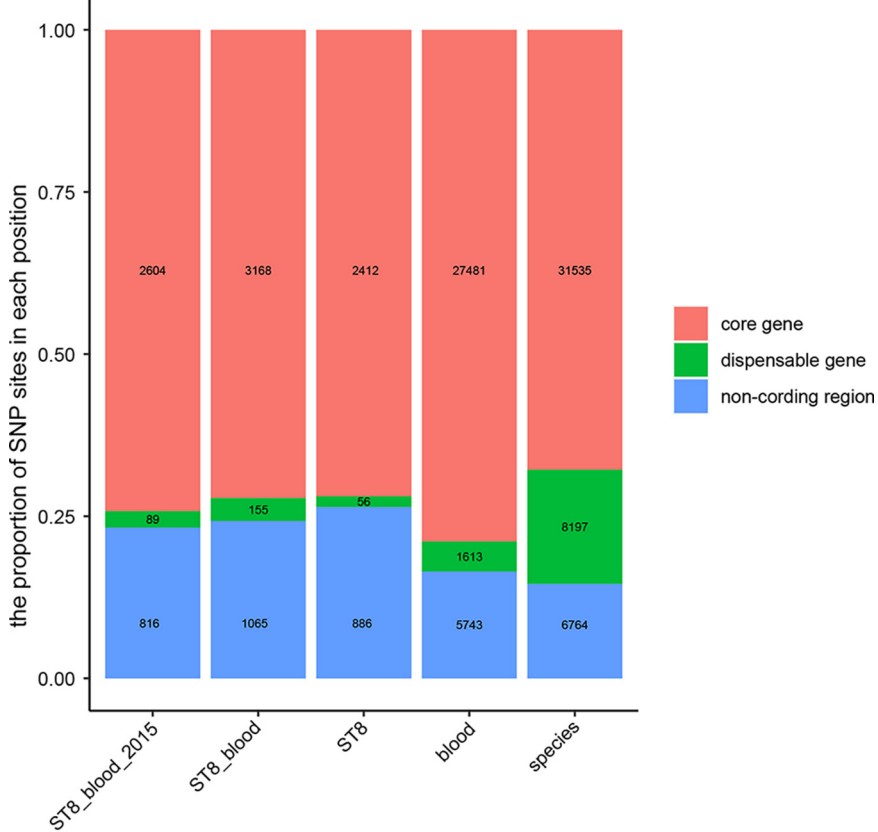

**FIG 5** Distribution of SNP sites in the reference genome. The number of SNP sites is marked in the stacked bar chart.

certain number of essential genes in this species. However, the number of core genes was not as high as previously thought (11–13). In the case of extreme strain diversity, we obtained a theoretical value that is relatively low in the number of core genes generally recognized in the literature. However, because the composition of bacterial genomes is affected by ST, source of isolation, date of collection, and other factors, we perform pan-genome analysis on strain populations to obtain pan-genome compositions that may be quite different from the theoretical values at the species level. Although the calculation of core genes (100%) is affected by genome integrity, we ensured the quality of the draft genomes used in this study by screening for ≤50 contigs. As the number of genomes increased, the number of 100% core genes decreased, suggesting that the genes required to ensure bacterial survival may involve a certain number of gene sets, while the deletion of individual genes in a strain did not necessarily affect its normal survival. Furthermore, the presence of many putative proteins in core genes suggests that our current understanding of essential life-sustaining functions is insufficient.

When we focused on the core genome, using different extrinsic conditions to control the strains revealed that the core genes were dynamically distributed among different strain populations. Overall, the strains analyzed in this study generally possessed similar characteristics, such as a specific environment and source of isolation. There were significant differences in dispensable genes and the distribution of core genes among different strain groups isolated in this way. Moreover, we cannot arbitrarily summarize the results of the pan-genome analysis of certain strains into the actual composition of the species' pan-genome when we carry out the pan-genome analysis of strains. Species-level representations of pan-genome composition are only possible if the strains included in the analysis are sufficiently diverse.

Among the four factors influencing gene dynamic mobility that were analyzed in this study, ST was the most important one. Furthermore, the local core gene region

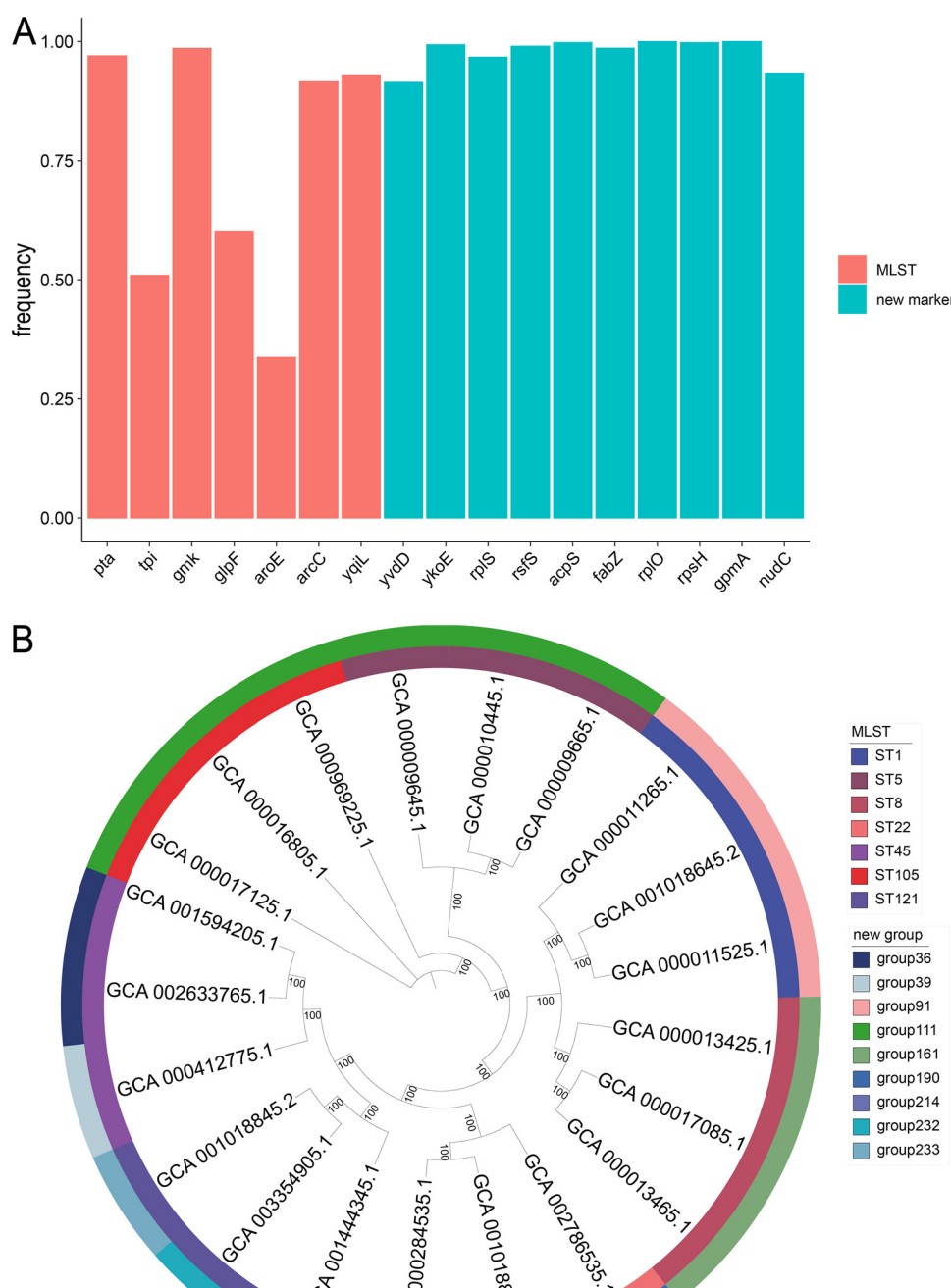

**FIG 6** Comparison of the frequency and classification between the new marker genes and the housekeeping genes. (A) Frequency of carrying 7 housekeeping genes and 10 new marker genes in 5,217 strains of *S. aureus*. (B) Twenty-one *S. aureus* strains were classified according to the two taxonomic methods and annotated in different colors.

shared by strains of the same ST suggests that we should pay more attention to the influence of evolutionary distance on gene distribution. Of the four factors, the one most frequently mentioned—i.e., isolation environment, including the impact of source on gene distribution—was not found to be dominant in this study. This may be because the *S. aureus* strains analyzed in this study are common pathogens that can survive in a variety of environments (14).

Currently, the most common method for epidemiological analysis in large-scale strain-screening work is whole-genome comparison to remove the influence of gene recombination and to use "core SNPs" for phylogenetic analysis (15, 16). However, the accuracy of the results

**TABLE 1** Information on 10 new marker genes

| Gene | No. of[a]: | | |
| --- | --- | --- | --- |
| | Nucleic acid sequence types | SNP sites | Length of nucleic acid sequence (bp) |
| acpS | 33 | 37 | 360 |
| fabZ | 40 | 44 | 441 |
| gpmA | 54 | 55 | 687 |
| nudC | 60 | 63 | 396 |
| rplO | 30 | 28 | 441 |
| rplS | 20 | 21 | 351 |
| rpsH | 19 | 16 | 399 |
| rsfS | 29 | 26 | 354 |
| ykoE | 67 | 76 | 576 |
| yvdD | 62 | 67 | 567 |

[a]There were 0 gap sites for each of the genes listed.

obtained from these methods is highly dependent on the integrity of the selected reference strains and the evolutionary distance between strains. If the strains included in the analysis originate from a wide range of sources, then the SNP sites on dispensable genes may affect the accuracy of the results, rendering this method of whole-genome alignment suitable for analyzing only strains with relatively close evolutionary distances, such as during short-term outbreak events (17, 18). When the strains included in the analysis come from a wide range of sources, multiple-sequence alignment of the core genes obtained by the pan-genome analysis should be used to assess evolutionary distance.

With the continuing increase in available *S. aureus* genomic data, the disadvantages of the housekeeping genes used in the current MLST method for *S. aureus* classification have become apparent. Approximately one-fifth of the strains discussed in this article did not contain all seven housekeeping genes, suggesting that the seven housekeeping genes selected earlier are no longer among the core genes of the species. The current MLST strategy is no longer universal for *S. aureus*. Although the 10 new marker genes we selected have not been verified via experiments and more advanced screening, they allow the classification of more strains. This highlights the necessity of finding a more suitable set of housekeeping genes for identification of *S. aureus* more effectively.

**Conclusion.** The results of our study clarified the distribution of theoretical values representing the number of *S. aureus* core genes and determined that these genes are dynamically distributed in different populations. Our analysis indicated that ST may be the primary factor driving the dynamic distribution of bacterial genomes, leading us to propose its potential impact on phylogenetic analysis. In addition, the new classification method proposed in this article suggests that we must find a set of housekeeping genes that are more suitable for *S. aureus* to improve the current classification status of this species.

## MATERIALS AND METHODS

**Genome data acquisition and basic analysis of *Staphylococcus aureus*.** The genomic sequencing data of the 5,217 *S. aureus* strains used in this study were downloaded from the NCBI database (https://ftp.ncbi.nlm.nih.gov/genomes/genbank/bacteria/Staphylococcus_aureus/); the assembly levels of each strain included complete genome, chromosome, and contig (number of contigs was ≤50) (see Data Set S1 in the supplemental material). We used the downloaded gff annotation files to extract metainformation for each strain and screen their source of isolation, country of isolation (geographic location), and date of collection. We reannotated the genomic files of bacterial genomes using Prokka, providing annotation files (e.g., gff3, gbk, and ffn) for subsequent genomic data analysis (19). To determine the MLSTs of all strains, we downloaded the nucleic acid sequences of the housekeeping genes and the profiles.csv file from the PubMLST website (https://pubmlst.org/data/). Then, we established a local BLAST database for all allele sequences of the seven housekeeping genes (*arc*, *aroE*, *glpF*, *gmk*, *pta*, *tpi*, and *yqiL*) and used blastn to compare the coding sequences of the strains with the local database of housekeeping genes (subject coverage = 100%; identity = 100%) according to profiles.csv to obtain the ST of each strain (20). We then used the automated software mlst (https://github.com/tseemann/mlst) again to reidentify the ST (21). We used Kraken2 and Bracken software for species annotation to identify the actual taxonomy of the strains with housekeeping gene deletions (22, 23).

**Pan-genomic analysis of *Staphylococcus aureus*.** Based on the gff3 annotation files output by Prokka, we used Roary for pan-genome analysis (24). Core genes were determined based on the gain-and-loss profiles of each gene in Roary's result file gene_presence_absence.csv. To calculate the theoretical number of core genes, all strains were randomly sampled 10 times in turn by a fixed number of strains to obtain the averages of 100, 99, and 95% of the number of core genes.

**Analysis of driving factors influencing the dynamic distribution of pan-genome.** We screened for strains containing complete information on ST and source of isolation, country, and year of collection. A least-squares regression model was built using LinearRegression in Python's sklearn module, with the number of strains containing the core gene ($y$) as the dependent variable and the independent variables as follows: (i) the number of STs with the gene present ($x_1$), (ii) the number of sources with the gene present ($x_2$), (iii) the number of countries with the gene present ($x_3$), and (iv) the number of years during which the gene was present ($x_4$). All data sets were analyzed by multiple regression. Hierarchical clustering of the gene-strain matrix was performed using the Heatmap package in R. The shell gene-strain matrix was obtained by screening the shell gene in the gene-strain matrix. (The proportion of strains carrying the gene ranged from 15% to 95%.) We used the vegan package in R to output the Jaccard distance matrix for the shell gene-strain matrix. We then performed a principal-component analysis (PCA) on the distance matrix and used K-means to perform unsupervised clustering based on two principal components (25).

**Genome-wide alignment and SNP calling.** We selected 5 groups (ST8_blood_2015, ST8_blood, ST8, blood, and species level) with 50 strains each and calculated the core genes (95%) in each group. We then selected a strain that existed in each group as a representative strain. Furthermore, the positions of the core gene region, dispensable gene region, and noncoding region in the genome of the reference strain were obtained according to the ID of the core gene and the gene position annotation file from Prokka. We used snippy (https://github .com/tseemann/snippy) to perform genome-wide alignment of the gbk file of the reference strain and the gff files of other strains to obtain the vcf file of the core gene SNP sites. The numbers of SNPs in the core gene, dispensable gene, and noncoding regions were counted according to SNP position and the position information for the reference genome.

**New classification of *Staphylococcus aureus* strains.** We screened genes according to the distribution of core genes (100%) in the distribution trend results of core genes and eliminated proteins with unknown functions and genes with multiple copies. We then used MUSCLE to perform multiple-sequence alignment between the alleles of screened genes (26). The genes with sequence lengths of >350 bp with no gap between alleles after multiple-sequence alignment were preserved. We assigned a sequence number (Data Set S2) to each allele of the newly determined marker and then assigned a group number to each allele combination as a new group.

**Gene frequency calculation and phylogenetic tree construction.** We selected a complete genome strain and used blastn to compare the ffn file (nucleotide coding regions fasta file) output by Prokka with the nucleic acid sequences of 7 housekeeping genes and 10 new marker genes to determine the annotation IDs of 17 genes in this strain. We then performed a pan-genomic analysis of the 5,217 strains using Roary, retrieving the annotation ID of the 17 genes in the selected strains in the results file gene_-presence_absence.csv to determine the row in which each gene was located and calculating the frequency of strains carrying each gene. We used the multiple-sequence alignment result file core_gene_a-lignment.aln in Roary's results as the FastTree input file for genome-wide phylogenetic analysis (27). We then used iTOL (https://itol.embl.de/) to visualize the phylogenetic tree (28).

**Data availability.** The data set analyses during the current study are available in the NCBI database.

## SUPPLEMENTAL MATERIAL

Supplemental material is available online only.

**SUPPLEMENTAL FILE 1**, PDF file, 5.3 MB.

**SUPPLEMENTAL FILE 2**, XLS file, 0.7 MB.

**SUPPLEMENTAL FILE 3**, XLS file, 0.4 MB.

**SUPPLEMENTAL FILE 4**, XLS file, 0.5 MB.

**SUPPLEMENTAL FILE 5**, XLS file, 0.6 MB.

## ACKNOWLEDGMENTS

This research was supported by the Fundamental Research Funds for the Central Universities with grant JKF-YG-22-B001.

Institutional review board and informed consent statements were not applicable to this study.

We declare no conflict of interest.

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
