## [Reviewer comments · Microbiology Spectrum]

Microbiology Spectrum

The pan-genome analysis of *Staphylococcus aureus* reveals key factors influencing genomic plasticity

Na Liu, Danping Liu, Kexin Li, Songnian Hu, and Zilong He

Corresponding Author(s): Zilong He, Beijing Advanced Innovation Center for Big Data-Based Precision Medicine, Interdisciplinary Innovation Institute of Medicine and Engineering, Beihang University

Review Timeline:

Submission Date:	August 16, 2022
Editorial Decision:	September 6, 2022
Revision Received:	September 26, 2022
Accepted:	October 13, 2022

Editor: Yanbin Yin

Reviewer(s): Disclosure of reviewer identity is with reference to reviewer comments included in decision letter(s). The following individuals involved in review of your submission have agreed to reveal their identity: Joao Carlos Gomes-Neto (Reviewer #3)

Transaction Report:

DOI: <https://doi.org/10.1128/spectrum.03117-22>

September 6, 2022

Prof. Zilong He

Beijing Advanced Innovation Center for Big Data-Based Precision Medicine, Interdisciplinary Innovation Institute of Medicine and Engineering, Beihang University
No. 37 Xueyuan Road, Haidian District,
Beijing 100191
China

Re: Spectrum03117-22 (The pan-genome analysis of Staphylococcus aureus reveals key factors influencing genomic plasticity)

Dear Prof. Zilong He:

Please consider both reviewers comments. If you are not able to perform the requested new analyses, please explain the reasons in the response letter.

Link Not Available

Sincerely,

Yanbin Yin

Journals Department
Reviewer comments:

Reviewer #1 (Comments for the Author):

The paper "The pan-genome analysis of Staphylococcus aureus reveals key factors influencing genomic plasticity" by Na Liu and colleagues performed pan-genome analysis using Staphylococcus aureus as the model organism and reveals the theoretical value of Staphylococcus aureus core genes and demonstrates from multiple perspectives that ST is a key influence on the plasticity of the Staphylococcus aureus genome. These works are of great significance in the field of bacterial pan-genome. In addition, they propose a new set of markers for MLST to suggest the shortcomings of the current MLST. The implication of the research is important and the methods are reasonable. However, authors should pay attention to some

details in the manuscript such as:

- 1) Line 220 should list the concrete content of x1-x4, although this part is presented in the methodology, it is necessary to list it again in the results section;
- 2) There are at least two errors in the application of quotation marks in the manuscript such as lines 229 and 239;
- 3) Please pay attention to the format of the text in the figures, for example, the font size of the figure title in Figure 4 is not uniform with other figures;
- 4) the references should pay attention to the format, such as the authors should be listed in full.

Reviewer #3 (Comments for the Author):

Dear authors,

I believe the work is relevant but needs some important considerations.

Starting from the perspective that this work look at variables that could contribute to the pan-genomic composition of the *S. aureus* population, and also address MLST classification accuracy, we need to carefully consider which algorithmic standards will be used for comparison and how generalizable the findings are toward other training datasets. Based on that here are my comments to improve the work:

1. As for the MLST accuracy and issues - I would recommend using ProkEvo, STing (<https://github.com/jordanlab/STing>), and mslt (<https://github.com/tseemann/mlst>), for comparison prior to determining how inaccurate the current mlst scheme is. I would also make sure that the type of data used is consistent with the platforms (typically it is Illumina paired-ended sequences). I would compare the accuracy and distribution of STs across all these platforms using the ~5,000 genomes prior to concluding that a new approach is needed. Once that testing data has been used, I would randomly grab another 5,000 from here (https://www.ncbi.nlm.nih.gov/pathogens/isolates/#taxgroup_name:%22Staphylococcus%20aureus%22) to compare accuracy and ST distribution, and the efficiency of your new approach.

2. Core-genes by roary - I don't see a need to change the threshold or even compare than to 99%. The population genetics definition of a core-gene would be 100% spread, but given errors in annotation, the 99% option is an optimal level.

3. From the initial testing data - please ascertain that all genomes are indeed *S. aureus*. That is why it is best to download genomes from here instead, where the metadata is more reliable.

https://www.ncbi.nlm.nih.gov/pathogens/isolates/#taxgroup_name:%22Staphylococcus%20aureus%22

4. I would use another approach to mine the pan-genomic data. How about using the shell-genes as defined by roary (hypothetical and annotated included) to identify clusters agnostically in the data using Jaccard distance and kmeans with the first 2 or 3 PCA of the distance, and then use that as a metric to associate with STs or other variables? And then ultimately you can look at patterns of divergence or convergence for the accessory-genome compared to the ST lineages. A logistic PCA coupled with kmeans could be used as well, but I would focus on using the shell-genes only. Or you could model the total accessory vs shell-genes. Once that is done and you include to the phylogeny that info, you could search for the differentiating annotated proteins that separate those groups/STs. Here is one example of such a work

<https://www.frontiersin.org/articles/10.3389/fsufs.2021.725791/full>

But I would use Jaccard distance from the vegan package instead of logistic PCA, and then would transform the distance matrix into PCs to use a k-means to identify clusters based on accessory-genome. Once that is established, you could then look at the core-genome regions linked to the accessory-genome clusters, if you so wish.

While doing this clustering analysis, you could use the PCs to be plotted and colored-coded based on other variables such as ST, and other epidemiological information. But my suggestion would be to do the modeling with your ~5,000 genomes, and then grab another completely independent sample of 5,000 genomes from separate SNP clusters present here

https://www.ncbi.nlm.nih.gov/pathogens/isolates/#taxgroup_name:%22Staphylococcus%20aureus%22

to demonstrate the validity.

Certainly this adds more work, but I believe it will be worth in the end, because if truly the MLST needs to be updated then it is worth demonstrating how in a very robust way.

Sincerely

Staff Comments:

Preparing Revision Guidelines

Please return the manuscript within 60 days; if you cannot complete the modification within this time period, please contact me. If you do not wish to modify the manuscript and prefer to submit it to another journal, please notify me of your decision immediately so that the manuscript may be formally withdrawn from consideration by Microbiology Spectrum.

Response to Reviewer Comments for Manuscript " The pan-genome analysis of *Staphylococcus aureus* reveals key factors influencing genomic plasticity" [Paper #Spectrum03117-22]

Dear Editors and Reviewers,

Thank you for your efforts in carefully reviewing our manuscript and the constructive suggestions that were provided. According to the reviewer's comments, we have revised the manuscript extensively, and a detailed list of the specific revisions is given below. These changes have been highlighted using yellow color. We hope that these changes can address your concerns and meet the requirements of publication. If there are any other modifications we could make, we would like very much to modify them and we really appreciate your help. We hope that our manuscript could be considered for publication in your journal. Thank you very much for your help. Again, we would like to express our sincere thanks for considering publishing our manuscript.

Reviewer #1:

The paper "The pan-genome analysis of *Staphylococcus aureus* reveals key factors influencing genomic plasticity" by Na Liu and colleagues performed pan-genome analysis using *Staphylococcus aureus* as the model organism and reveals the theoretical value of *Staphylococcus aureus* core genes and demonstrates from multiple perspectives that ST is a key influence on the plasticity of the *Staphylococcus aureus* genome. These works are of great significance in the field of bacterial pan-genome. In addition, they propose a new set of markers for MLST to suggest the shortcomings of the current MLST.

Response: Thank you for your comments on our manuscript. We appreciate your clear and detailed suggestion. We have revised the corresponding part of the article according to your suggestion. We hope the following explanation can address your concerns.

The implication of the research is important and the methods are reasonable. However, authors should pay attention to some details in the manuscript such as:

1) Line 220 should list the concrete content of x_1-x_4 , although this part is presented in the methodology, it is necessary to list it again in the results section;

Response: Thank you for the suggestion. According to your suggestion, we think it is necessary to re-list the specific meaning of x_1-x_4 in the result section, and the corresponding content has been added in line 225-227.

2) There are at least two errors in the application of quotation marks in the manuscript such as lines 229 and 239;

Response: Thanks for your comments. We have fixed the two quotation marks errors you raised (now at line 236 and line 237). And we have checked and corrected the quotation marks throughout the text.

3) Please pay attention to the format of the text in the figures, for example, the font size of the figure title in Figure 4 is not uniform with other figures;

Response: Thanks for your reminding. We have changed the font in the figure (this figure is labeled as figure 5 in the revised manuscript) and checked the fonts in the other figures. Thank you again for your suggestions on the details of our manuscript.

4) the references should pay attention to the format, such as the authors should be listed in full.

Response: Thank you for your reminder. We downloaded the "ASM Journals.ens" file from the journal's homepage to make modifications to the reference format to meet publication requirements.

Reviewer #3:

I believe the work is relevant but needs some important considerations.

Starting from the perspective that this work look at variables that could contribute to the pan-genomic composition of the *S. aureus* population, and also address MLST

classification accuracy, we need to carefully consider which algorithmic standards will be used for comparison and how generalizable the findings are toward other training datasets. Based on that here are my comments to improve the work:

Response: Thanks for your comments on our manuscript. We appreciate your clear and detailed feedback, and we have tried our best to follow your useful suggestions as well as we could. We did more analysis works according to your suggestion. We hope the following explanation can address your concerns.

1. As for the MLST accuracy and issues - I would recommend using ProkEvo, STing (<https://github.com/jordanlab/STing>), and mslst (<https://github.com/tseemann/mlst>), for comparison prior to determining how inaccurate the current mlst scheme is. I would also make sure that the type of data used is consistent with the platforms (typically it is Illumina paired-ended sequences). I would compare the accuracy and distribution of STs across all these platforms using the ~5,000 genomes prior to concluding that a new approach is needed. Once that testing data has been used, I would randomly grab another 5,000 from here (https://www.ncbi.nlm.nih.gov/pathogens/isolates/#taxgroup_name:%22Staphylococcus%20aureus%22) to compare accuracy and ST distribution, and the efficiency of your new approach.

Response: Thank you very much for your suggestion. Following your advice, we have reviewed the documentation for the software ProkEvo, STing, and mslst. We have found that ProkEvo is an excellent pipeline for bacterial genome analysis and pan-genomic studies. Multilocus-sequence typing is also performed on the assemblies using mlst in ProkEvo. As the data in this study were assemblies, we validated the strain ST using mlst and cited both ProkEvo and mlst. This part was put to the methods section as indicated in line 125-126. Comparing the identification results we found that some of the strains initially identified as ST unknown were not accurately identified due to the limitations of our parameter settings. Based on the validation that 216 of the 5217 strains used in this study were typed as unknown. This part of the results has been revised in line 184-186. Based on the above results, we consider your proposed validation regarding the accuracy of the MLST protocol and the necessity of

finding new marker genes. We calculated the frequency of the seven housekeeping genes and the new marker genes carried in the strains by analyzing the orthologs in the Roary results for each strain. We found that some of the housekeeping genes were not core genes in strain 5217 *Staphylococcus aureus* (Fig 6A). At the same time, these newly screened marker genes have better performance (*rsmD* has been removed from the new marker list because it is carried less than 90% of the time, and any changes needed in the corresponding results have been updated and marked in yellow in line 285-298 and line 352). At your suggestion, we randomly selected additional 5289 strains of *S. aureus* (https://www.ncbi.nlm.nih.gov/pathogens/isolates/#taxgroup_name:%22Staphylococcus%20aureus%22) for validation of the 17 gene carriage frequencies, and the low carriage frequency of the housekeeping gene *arcC* also supports the idea that a new set of markers needs to be found. The methods, results, and figure legends of this section are added to line 171-177, line 275-284, and line 495-497.

2. Core-genes by roary - I don't see a need to change the threshold or even compare than to 99%. The population genetics definition of a core-gene would be 100% spread, but given errors in annotation, the 99% option is an optimal level.

Response: Thanks for your reminder. We strongly agree with you that 99% could already indicate the distribution of the number of core genes in *S. aureus*. Unknown errors in strain genome quality, and genome annotation may indeed affect 100% of the core genes. The fact that the genomes of the strains included in this part are complete, chromosome or contigs ≤ 50 assembly level, and that the strains have passed a rigorous species identification has ensured that the potential risk of error in the calculation of 100% core genes is minimal in terms of genome quality. As you say the definition of core genes in population genetics is 100% transmission, and 100% core genes are of some reference value while maintaining the quality of the genomic data. We therefore decided to retain the results of the core gene (100%) calculations. Thank you again for valuable comments.

3. From the initial testing data - please ascertain that all genomes are indeed *S. aureus*. That is why it is best to download genomes from here instead, where the metadata is more reliable.
https://www.ncbi.nlm.nih.gov/pathogens/isolates/#taxgroup_name:%22Staphylococcus%20aureus%22.

Response: Thank you for your advice. Thank you for providing us with a link to download more reliable metadata of strains. We download the "isolates.tsv" file containing the strain metadata at the link you suggested. A search of the GCA numbers of the 5217 strains used in this article revealed that a total of 5136 strains exist in "isolate.tsv". An additional 81 strains were identified as *Staphylococcus aureus* by the species information on the download site and by krakren2 species identification, indicating that the strains we used are reliable. And we downloaded 5289 verified strains from https://www.ncbi.nlm.nih.gov/pathogens/isolates/#taxgroup_name:%22Staphylococcus%20aureus%22. Thank you very much for your reminder.

4. I would use another approach to mine the pan-genomic data. How about using the shell-genes as defined by roary (hypothetical and annotated included) to identify clusters agnostically in the data using Jaccard distance and kmeans with the first 2 or 3 PCA of the distance, and then use that as a metric to associate with STs or other variables? And then ultimately you can look at patterns of divergence or convergence for the accessory-genome compared to the ST lineages. A logistic PCA coupled with kmeans could be used as well, but I would focus on using the shell-genes only. Or you could model the total accessory vs shell-genes. Once that is done and you include to the phylogeny that info, you could search for the differentiating annotated proteins that separate those groups/STs. Here is one example of such a work <https://www.frontiersin.org/articles/10.3389/fsufs.2021.725791/full> But I would use Jaccard distance from the vegan package instead of logistic PCA, and then would transform the distance matrix into PCs to use a k-means to identify clusters based on accessory-genome. Once that is established, you could then look at the core-genome

regions linked to the accessory-genome clusters, if you so wish. While doing this clustering analysis, you could use the PCs to be plotted and colored-coded based on other variables such as ST, and other epidemiological information. But my suggestion would be to do the modeling with your ~5,000 genomes, and then grab another completely independent sample of 5,000 genomes from separate SNP clusters present here

https://www.ncbi.nlm.nih.gov/pathogens/isolates/#taxgroup_name:%22Staphylococcus%20aureus%22 to demonstrate the validity.

Response: We are extremely grateful for pointing out this problem. We refer to the research you mentioned for the following analysis: we analyzed shell genes using Jaccard distance and PCA according to your suggestion and performed unsupervised clustering using K-means based on the first two principal components. The clusters obtained by K-means and ST were used to show the clustering of strains, respectively. This validated our proposed ST as a key factor influencing the genomic plasticity of the strains. The methods, results, and figure legends of this section are added to line 144-149, line 250-256, and line 485-492. Thank you very much for your advice and for providing us with a reference, and we have included this article as our reference (no. 17). In addition, the section on "model the total accessory vs shell-genes" that you mentioned is our next work. As you mentioned, we will focus on the differential proteins in different ST strains, which are important for the pan-genome. In this article, we mainly demonstrate that ST is an important influence on genomic plasticity. The articles you mention are also of methodological value for our subsequent research. Thank you again for your valuable comments on this study.

Certainly this adds more work, but I believe it will be worth in the end, because if truly the MLST needs to be updated then it is worth demonstrating how in a very robust way.

Response: Thank you again for your comments and useful suggestions on our manuscript. We explain your questions about the validation of the MLST as well as in

the corresponding section earlier. We hope that our work has addressed the questions you have raised about this study.

October 13, 2022

Prof. Zilong He

Beijing Advanced Innovation Center for Big Data-Based Precision Medicine, Interdisciplinary Innovation Institute of Medicine and Engineering, Beihang University
No. 37 Xueyuan Road, Haidian District,
Beijing 100191
China

Re: Spectrum03117-22R1 (The pan-genome analysis of *Staphylococcus aureus* reveals key factors influencing genomic plasticity)

Dear Prof. Zilong He:

Your manuscript has been accepted, and I am forwarding it to the ASM Journals Department for publication. You will be notified when your proofs are ready to be viewed.

Sincerely,

Yanbin Yin
Editor, Microbiology Spectrum

Journals Department
Supplemental Dataset S1: Accept
Supplemental Dataset S3: Accept
Supplemental Material: Accept
Supplemental Dataset S2: Accept
Supplemental Dataset S4: Accept